# Pathogenesis of Port-Wine Stains: Directions for Future Therapies

**DOI:** 10.3390/ijms232012139

**Published:** 2022-10-12

**Authors:** Lian Liu, Xiaoxue Li, Qian Zhao, Lihua Yang, Xian Jiang

**Affiliations:** 1Department of Dermatology, West China Hospital, Sichuan University, Chengdu 610017, China; 2Laboratory of Dermatology, Clinical Institute of Inflammation and Immunology (CIII), Frontiers Science Center for Disease-Related Molecular Network, West China Hospital, Sichuan University, Chengdu 610017, China; 3Department of Medical Cosmetology, Chengdu Second People’s Hospital, Chengdu 610056, China

**Keywords:** port-wine stain, pathogenesis, treatment

## Abstract

Port-wine stains (PWSs) are congenital vascular malformations that involve the skin and mucosa. To date, the mechanisms underlying the pathogenesis and progression of PWSs are yet to be clearly elucidated. The potential reasons for dilated vessels are as follows: (1) somatic *GNAQ* (R183Q) mutations that form enlarged capillary malformation-like vessels through angiopoietin-2, (2) decreased perivascular nerve elements, (3) the coexistence of Eph receptor B1 and ephrin B2, and (4) the deficiency of αSMA expression in pericytes. In addition, ERK, c-JNK, P70S6K, AKT, PI3K, and PKC are assumed to be involved in PWS development. Although pulsed-dye laser (PDL) remains the gold standard for treating PWSs, the recurrence rate is high. Topical drugs, including imiquimod, axitinib, and rapamycin, combined with PDL treatments, are expected to alter the recurrence rate and reduce the number of PDL sessions for PWSs. For the deep vascular plexus, photosensitizers or photothermal transduction agents encapsulated by nanocarriers conjugated to surface markers (CD133/CD166/VEGFR-2) possess a promising therapeutic potential in photodynamic therapy or photothermal therapy for PWSs. The pathogenesis, progression, and treatment of PWSs should be extensively investigated.

## 1. Definition and Epidemiology of PWSs

Port-wine stains (PWSs) are congenital vascular malformations characterized by ectasia of capillary and postcapillary venules in the skin and mucosa. The incidence of PWSs in newborns is approximately 0.3% to 0.5%, and they affect about 25 million people worldwide [1,2,3]. The lesions primarily manifest as flat and pink to red irregular patches at birth. Most of these lesions gradually develop into hypertrophic, nodular, and darker lesions over time.

A retrospective study analyzed 425 patients with facial PWSs and reported that 65% of all patients with PWSs developed hypertrophic or nodular lesions in the fifth decade of life. In addition, the mean age of hypertrophy is noted to be 37 years old [4]. Similarly, another study revealed that 60% of patients (96/160) with PWSs had soft tissue/bony hypertrophy or nodule formation by the age of 9 years old. Moreover, nodules appeared in 38/87 (44%) of the patients, with an average age of onset of 22 years old [5]. A previous study found that 45% of the lesions appeared to be restricted to one of three areas of the sensory trigeminal nerve distribution, while 55% were involved in multiple dermatomes or crossed the midline [6]. In addition, the mucous membranes are often contiguously affected. In a study of 310 patients with PWSs, 68% of the patients had more than one dermatome affected, 85% had unilateral distribution, and 15% had bilateral distribution [7]. Because 70% to 80% of the lesions occur on the face and neck, PWSs can negatively affect the psychosocial development and well-being of patients due to the consciousness of their abnormal cosmetic appearance [8,9].

PWSs can occur alone or as part of vascular syndromes, including Sturge–Weber syndrome (SWS), Klippel–Trenaunay syndrome (KTS), phakomatosis pigmentovascularis (PPV), and Proteus syndrome. Patients with such syndromes can develop glaucoma, arteriovenous malformations, arteriovenous fistulas of limbs, leptomeningeal vascular malformations, etc. Color Doppler ultrasound, three-dimensional CT angiography, and enhanced magnetic resonance imaging are needed to confirm these diseases. Moreover, patients should be assessed by a multidisciplinary team (MDT) with different specialties involving several areas, including dermatology, pediatric, neurology, otolaryngology, radiology, neurosurgery, and ophthalmology.

The treatment of PWSs is still a challenge worldwide, even though recent advancements have been made in treatment [10]. The effectiveness of treatment can be affected by multiple factors, such as age, the location of lesions, and skin color [11,12]. It is substantially associated with the characteristics of vessels, such as vessel diameter, depth, density, and wall thickness. Furthermore, the classification of PWSs can partly reflect the characteristics of vessels. In PWSs, lesions can be classified as flat, hypertrophic, and nodular, as well as pink, red, purple, dark red, or dark purple, based on their thickness, shape, and color [13,14,15,16]. A randomized controlled trial revealed that the patients who had pink-type PWSs responded better to the photodynamic therapy (PDT)–hemoporfin treatment [17]. Moreover, 1064 nm Nd: YAG lasers and 755 nm lasers have been used for the treatment of hypertrophic PWSs, which is highly effective [18,19,20]. However, no internationally accepted criteria have been made for the classification of PWSs. Therefore, PWSs should be properly classified.

## 2. Pathogenesis

### 2.1. Pathological Characteristics of PWSs

#### 2.1.1. The Vessel Types and Evolution of PWSs

PWSs are generally considered to be capillary malformations (CMs) [21]. However, new perspectives on the vessel types of PWSs are emerging. As a result of the staining properties of the PAL-E antibody, the vessels of PWSs are classified as capillaries, postcapillary venules, and small veins [22]. PWSs are progressive diseases, and their pathologic characteristics may change over time. A previous study showed that typical PWSs are characterized by the proliferation and dilation of thin-walled vessels in the dermis and subcutaneous tissue (cavernous hemangioma), and the nodules in the PWSs are composed of variable-thickness vessels and intervascular stroma (arteriovenous malformations (AVMs)) [23]. Similarly, another study focused on the pathological characteristics of nodules in PWSs. Among the biopsy specimens, 14 were classified as pyogenic granulomas (PGs) (45.1%), 10 as AVMs (32.3%), 5 as both (16.1%), and 2 as cavernous vascular ectasia (6.5%) [24]. To understand the progression of PWSs better, we categorized PWSs into the following four groups: red PWSs, purple PWSs, hypertrophic PWSs, and nodular PWSs. We then investigated their pathologic features. The vessels in red PWSs, purple PWSs, and hypertrophic PWSs were mainly composed of CMs and venous malformations (VMs), whereas those in nodular PWSs were either VMs or AVMs [25]. To summarize, PWSs are initially characterized as capillaries and/or postcapillary venous malformations, and the nodules in PWSs can be VMs, AVMs, PGs, or cavernous vascular ectasia.

#### 2.1.2. The Characteristics of Endothelial Cells, Pericytes, and Basement Membranes

Capillaries are composed of endothelial cells (ECs), pericytes, and basement membranes (BMs) with tight junctions. BMs are mainly composed of laminins, type IV collagen, perlecan, and nidogen. Therefore, previous research has studied ECs, pericytes, and BMs to elucidate the pathogenesis of PWSs. A comparison was performed of the distribution of factor VIII (as a marker for ECs) and type IV collagen (a component of BMs) in normal skin and PWSs via immunofluorescent staining [26]. The location, character, and intensity of factor VIII and type IV collagen showed no difference between normal skin and PWSs. Therefore, the ectasia of vessels was caused by structural support change rather than an abnormality of ECs and BMs.

Another study further investigated the characteristics of the ECs and BMs in PWSs. ECs swelled or flattened in response to mild or pronounced dilatation of vessels, while the markers of BMs (laminin, type IV collagen, and fibronectin) were stained more extensively [21]. These results further confirmed that PWSs are CMs rather than hemangiomas. Moreover, the distribution and staining patterns of PAL-E, ICAM-1, ELAM-1, and FVIIIrAg, which recognize specific epitopes on vascular ECs, were investigated [22]. No significant difference was found in the intensity and distribution pattern of these proteins between normal skin and PWSs, which was consistent with previous findings. Therefore, these results seemed to show that abnormal vessels are not the result of endothelial defects. However, a recent study revealed that ECs are differentiation-impaired, and the late-stage progenitors of the endothelium form immature venule-like vessels [27]. Moreover, the coexistence of Eph receptor B1 and ephrin B2 disrupts the normal interactions between ECs, which results in the progressive dilatation of vessels in PWSs (Figure 1).

Pericytes, also known as mural cells or Rouget cells, are embedded in BMs and envelop the ECs in vessels. Along with forming and stabilizing blood vessels, pericytes can regulate the diameter and blood flow of the capillary [28,29]. Moreover, contractile pericytes express α smooth muscle actin (αSMA) at the arteriole end of the capillary bed [30]. A deficiency of αSMA expression was observed in the pericytes from dilated PWS vessels, compared with normal vessels [31]. Therefore, it is also possible that abnormal αSMA expression in pericytes contributes to the progressive dilatation of PWS blood vessels. Presently, only one study has observed pericytes in PWSs, and further research is required to confirm the scientific validity of this hypothesis.

#### 2.1.3. The Ultrastructural Characterization of PWSs

A study showed that the wall material of vessels is primarily composed of amorphous deposits interspersed with collagen fibrils, and the amount of the basement membrane is occasionally increased [32]. Furthermore, most vessels were found to have fenestrations and/or small gaps on the endothelium, and cross-banded filamentous aggregates were observed around the walls. Therefore, it was suggested that the dilatation of the vessels is associated with structural changes in the vascular and intervascular connective tissues. Moreover, the stability and permeability of the endothelium can be affected. In another study, the layers of pericytes and BM were increased in infantile and early childhood PWSs [31]. In addition, the disorganization and hypertrophy of collagen and elastic fibers were observed, along with the degeneration of smooth muscles. According to these findings, the entire physiological milieu of the skin is affected by PWSs.

Almost 65% of patients with PWSs present hypertrophic or nodular lesions by the fifth decade of life. Hence, the ultrastructure of hypertrophic or nodular lesions was investigated. Compared with normal skin, ECs, pericytes, and fibroblasts contain abundant organelles, which include the Golgi apparatus, the endoplasmic reticulum, and mitochondria [33]. Furthermore, hyperactive fibroblasts increase the number of collagenous bundles under the epidermis. In addition, hypertrophic and nodular PWS specimens showed hobnailed ECs with gaps between them. Such gaps between ECs have previously been reported [32], and specific therapeutic materials such as nanomaterials may benefit from the ultrastructure.

### 2.2. Gene Mutations

Even though the majority of PWS lesions are sporadic, without gender differences, and typical familial cases are uncommon, there are controversies regarding the hereditary nature of the disorder. Some studies have suggested that PWSs are caused by somatic mutations or post-zygotic mutations. In two pairs of monozygotic twins, one of each twin developed PWSs [34]. As both pairs were monochorionic diamniotic twins, post-zygotic events may be involved in the pathogenesis of PWSs.

#### 2.2.1. The Inheritance of PWS

PWSs are usually sporadic, and the occurrence of PWSs in families is relatively rare. However, in a prospective study, 7.8% of patients (22/283) with PWSs had a family history [35]. Moreover, a retrospective study found that 27% (65/240) of patients with PWSs had a hereditary condition [36]. Furthermore, four subjects with classical PWSs were present in a family. II.2 and II.4 were sisters (Appendix A), and both of them had fifth finger clinodactyly, without PWSs. III.1, III.3, and III.4 had typical PWSs and fifth finger clinodactyly, whereas III.2 only had PWSs without fifth finger clinodactyly [37]. It was suggested that PWSs are inherited as monogenic disorders, with reduced penetrance and X-linked inheritance, consistent with an autosomal dominant trait. Therefore, scholars explored the possible gene that could influence heredity.

To better elucidate the genetic mechanism of inherited PWSs, researchers have performed a genome-wide linkage study on six families with inherited PWSs [38]. They found the cosegregation of PWSs with a large locus on chromosome 5q. Several genes involved in vascular and neural development were located in this locus, which included *MEF2C*, *RASA1*, and *THBS4*. Capillary malformation–arteriovenous malformation (CM–AVM) is known to be caused by mutations in *RASA1*. A study investigated whether *RASA1* mutation is responsible for familial PWSs [36]. The blood of 65 patients who had a positive family history of PWSs was collected. Interestingly, none of the patients were found to have *RASA1* gene mutations via denaturing high-performance liquid chromatography. There are several possible reasons for the negative results, including the following: (1) inappropriate detection techniques, (2) the incorrect selection of the gene; (3) PWSs are not a monogenic disorder. Despite the absence of positive results in the past, research regarding familial PWSs is still required as experimental techniques advance.

#### 2.2.2. GNAQ Mutations

Shirley et al. first reported in 2013 that PWSs and SWS were associated with the somatic *GNAQ* mutation (c.548G > A, p.Arg183Gln, R183Q) [39]. A majority of patients (12/13) with PWSs had *GNAQ* mutations detected by SNaPshot assays. Furthermore, *GNAQ* (c.548G > A) mutations modestly increased the activity of extracellular signal-regulated kinase (ERK).

It is well known that PWSs are vascular disorders. Hence, some scholars have wondered whether *GNAQ* (R183Q) mutations in PWSs are enriched in blood vessels or other skin tissues. By combining laser capture microscopy (LCM) with next-generation sequencing, Tan, W. et al. detected *GNAQ* (R183Q) mutations in four different types of skin structures, which included blood vessels, hair follicles/glands, connective tissues, and the epidermis [40]. The study revealed that *GNAQ* (R183Q) mutations were primarily found in blood vessels in 6 out of 10 patients, with mutation frequencies ranging from 3.16% to 12.37%. Furthermore, another study also demonstrated that somatic *GNAQ* (R183Q) mutations were enriched in the ECs from CMs, using fluorescence-activated cell sorting (FACS) and droplet digital PCR (ddPCR) [41]. Additionally, Ma, G. et al. investigated the frequencies of *GNAQ* (R183Q) mutations in the different tissues of patients with port-wine macrocheilia (PWM) exhibiting primary hypertrophy [42]. Next-generation targeted sequencing revealed that *GNAQ* (R183Q) mutations were found in 90% of the skin samples, 90% of the mucosal samples, and 90% of the muscle samples; however, it was not detected in the gland samples.

A study identified that *GNAQ* (R183Q) mutations were more common in individuals with PWSs involving all three facial regions (upper, middle, and lower) than in individuals with PWSs involving just one or two [43]. Additionally, the patients with PWSs involving extremities showed a lower rate of *GNAQ* (R183Q) mutations than those with PWSs affecting the head and neck. Therefore, clinical phenotypes were actually associated with the mutations of *GNAQ* (R183Q). Researchers also discovered a new mutation of *GNAQ* (g.9: 80412494G > C, p.R183G) in a Chinese patient with PWSs [44]. Overall, the pathogenesis of PWSs was closely associated with *GNAQ* mutations, with the mutation frequency ranging from 55.6% to 100% [41,45].

The role of GNAQ (R183Q) mutations in the pathogenesis of PWSs is still questionable. The *GNAQ* gene, which encodes the Gαq protein, is located on chromosome 9q21.2 [46]. Gαq contains an α helical domain and a Ras-like GTPase domain, which can form a pocket to bind the guanosine diphosphate (GDP) [47]. G-protein-coupled receptors (GPCRs) are signal-conveying proteins that play important role in the development of sensory functions, normal organ functions, and neurologic functions [48,49,50]. In response to cognate ligand binding, GPCRs bind the G proteins and catalyze the dissociation of GDP from Gα for GTP. Once Gα binds GTP, the dissociation of the Gβγ subunit from the Gα subunit is triggered. Signaling cascades are activated, and biological processes are regulated when active GTP-α and βγ complexes are present.

Previous studies showed that the substitution of cysteine at R183 stabilized the inactive GDP-bound Gαq conformation and reduced the hydrolysis of GTP to GDP (Figure 1) [51]. One such study revealed that *GNAQ* (R183Q) mutations modestly increased the activity of ERK. To further elucidate the underlying mechanism of *GNAQ* (R183Q) mutations, experiments on these types of mutations were performed in vitro and in vivo [52]. Endothelial colony-forming cell lines were transfected with lentivirus expressing p.R183Q *GNAQ* (EC-R183Q). Based on bulk RNA sequencing, protein kinase C (PKC), nuclear factor kappa B, and calcineurin signals were activated, and angiopoietin-2 (ANGPT2) was increased. After injecting EC-R183Q in combination with bone marrow mesenchymal progenitor cells under the skin of nu/nu mice, enlarged blood vessels that resembled CMs were formed. Furthermore, the suppression of ANGPT2 could prevent the enlargement of vessels.

#### 2.2.3. Other Gene Mutations

Even though *GNAQ* (R183Q) mutations were mainly identified in PWSs, other gene mutations can also participate in the pathogenesis of PWSs. Targeted next-generation sequencing for 275 known cancer genes was performed in PWSs and normal skin. *GNAQ* (R183Q) mutations were found as expected; however, novel mutations were also discovered, such as *PIK3CA*, *SMARCA4*, *EPHA3*, *MYB*, and *PDGFR-β* [53]. The *PIK3CA* gene encodes the p110⍺ catalytic subunit of PI3K, and it plays a critical role in PI3K-related progression. The mutations of *PIK3CA* are associated with the activation of the PI3K–AKT pathway, which regulates angiogenesis [54]. Moreover, *PIK3CA* mutations were identified in CLOVES and lymphatic anomalies [55,56]. This suggests that *PIK3CA* mutations are related to vascular anomalies, including CMs. *SMARCA4* (also known as *BRG1*) is a chromatin-remodeling complex that plays a role in the development of embryonic vessels, specifically promoting venous differentiation [57,58]. The loss of *BRG1* is associated with poor vessel remodeling, blunted ends, or non-interconnected vessels. The ephrin/Eph system is the largest family of tyrosine kinase receptors and is involved in vasculogenesis, vascular remodeling, angiogenesis, axon guidance, and synaptic plasticity [59]. Cell adhesion, repulsion, and motility are mediated by EphA3, which may be related to the ectatic vascular structure in PWSs. Although researchers have found some novel mutations that might be associated with PWSs, these genes were not verified via first-generation sequencing or screened for functional verification. Therefore, further study is required.

### 2.3. Aberrant Activation of the Kinase

According to the literature, *GNAQ* (R183Q) mutations are closely associated with PWSs, with a mutation frequency of 55.6% to 100%. The mutations of *GNAQ* (R183Q) can increase the activation of downstream signaling, which results in the modest activation of ERK. In fact, Gαq proteins encoded by *GNAQ* are associated with the PI3K/AKT, AMPK, MAPK, and mTOR signaling pathways. Hence, a study investigated the activation status of various kinases in PWS lesions, including ERK, c-JNK, AKT, PI3K, P70S6K, and PLC-γ [60]. It showed that ERK, c-JNK, and P70S6K were activated in the PWS vessels of children and adults. In many adults, hypertrophic PWS blood vessels, AKT, and PI3K were activated but not in infants. Meanwhile, PLC-γ was strongly activated in nodular PWS blood vessels. Moreover, another study analyzed kinases and proteins in hypertrophic and nodular lesions [61]. Compared with the adjacent normal skin, the phosphorylated levels of PKCα, PI3K, PDPK1, and PLC-γ, as well as protein levels of PP2A and DAG, were moderately increased in the ECs of hypertrophic PWSs. However, they were strongly increased in the vasculatures and surrounding fibroblasts in PWS nodules. In summary, c-JNK and ERK contribute to the progression and pathogenesis of PWSs, as they are first activated in all PWS tissues, and AKT and PI3K are subsequently activated, which are involved in the development of hypertrophic PWSs. PKCα and PI3K signaling pathways contribute to the development of hypertrophic and nodular PWSs.

### 2.4. Upregulation of Membrane Trafficking and Exocytosis

In recent years, a variety of omic technologies have been developed, including transcriptomics, proteomics, and metabolomics. Although numerous studies revealed that *GNAQ* mutations were related to PWSs using whole-exome sequencing and next-generation sequencing, few studies focused on the transcriptomics and proteomics of PWSs.

PWS lesions were identified via the sequential windowing acquisition of all theoretical fragment ion mass spectra (SWATH-MS) [62]. Compared with normal skin, 107 out of the 299 identified proteins were differentially expressed in PWS lesions and were mainly related to membrane trafficking, cytoskeleton, and collagen. According to confirmatory studies, membrane trafficking/exocytosis-related proteins were significantly increased, and extracellular vesicle exocytosis was upregulated in vessels observed by using transmission electron microscopy. Hence, these results suggested that ECs release extracellular vesicles that might act as intercellular signaling mediators in the pathogenesis of PWSs. Although membrane trafficking and exocytosis were associated with the pathogenesis of PWSs, more meaningful proteins were not found. Several reasons may explain this: (1) the sample size is small (*n* = 6), (2) proteome detection techniques are not advanced, and (3) PWSs are classified as diseases affecting the entire skin; however, gene mutations are predominantly located in ECs. Thus, flow cytometry can be used to distinguish different cell types for detection by using proteomics. In addition, spatial proteomics may also help analyze the proteins from PWS lesions.

### 2.5. Reduction in Neural Innervation around the Ectatic Blood Vessels

As is well known, nerves can influence the contraction and dilation of vessels, and both sympathetic and sensory nerve endings can innervate vessels in the skin. A study of 310 patients with PWSs revealed that 68% had more than one dermatome affected, 85% had unilateral distributions, and 15% had bilateral distributions [7]. Furthermore, PWSs usually coincide with the distribution of the trigeminal nerve branches. Hence, some scholars speculated that the pathogenesis of PWSs was associated with the abnormal neural regulation of blood flow.

Scholars performed studies on the density and function of nerves in the dermis to prove the above hypothesis. S100 is the special marker of myelinating or non-myelinating Schwann cells and Meissner corpuscle capsules [63]. Therefore, immunoperoxidase staining with S100 is used to confirm nerve density. In PWSs, only 17% ± 3% of vessels are associated with nerves; however, 75% ± 11% of vessels course within nerves in normal skin. The study revealed that the nerve density of PWSs was significantly reduced, and the vessel-to-nerve ratio was remarkably increased, compared with those of normal skin. The decrease in perivascular nerve elements in PWSs may alter the neural modulation of vascular tone, further affecting progressive vascular ectasia.

Generally, sympathetic nerves are considered to be responsible for controlling the contraction and relaxation of skin blood vessels, but sensory nerve fibers may also be involved. A study used the neuronal cytoplasmic protein gene product 9.5 (PGP9.5) and neuron-specific enolase (NSE) to label total innervation in the skin, and neurofilament (NF) proteins and neuropeptide calcitonin gene-related peptide (CGRP) to mark sensory nerve fibers [64]. Surprisingly, no or only occasional nerve fibers (total innervation and sensory nerve fibers) were observed around ectatic vessels, whereas nerve fibers with normal morphology were found in other skin structures. Selim, M.M. et al. also demonstrated that the nerve density was reduced, and the vascular density and the mean vessel diameter increased in those patients with PWSs who did not respond to treatment [65]. Moreover, the function of blood vessels was detected with vasodilator cream and vasoconstrictor steroid tape [66]. Changes in blood flow after the application of the cream and tape were significantly reduced in PWSs than in normal skin. These results indicated that the vascular stimulation to vasoactive substances in PWSs was weakened, suggesting the lack of vascular innervation in PWSs.

Although studies have shown that PWSs are associated with neurological deficits, the cause of neurological deficits is still unclear. Some scholars speculate that the defective neurotrophic factors during embryogenesis are related to focal nerve deficits [67]. According to recent studies, *GNAQ* (R183Q) mutations are associated with the pathogenesis of PWSs; thus, these gene mutations could also be associated with neurological deficits. However, further studies are needed to confirm the relationship.

## 3. Treatment of PWSs: Current Challenges and Future Perspectives

### 3.1. The Current Challenges in the Treatment of PWSs: Recurrence and Resistance

Before the theory of selective photothermolysis was proposed, PWSs were treated with tattooing, cryosurgery, ionizing radiation, CO_2_ lasers, and electrotherapy, with low effectiveness and significant side effects [68,69,70]. In 1983, Anderson and Parrish introduced the theory of selective photothermolysis, revolutionizing the treatment of PWSs [71]. Currently, pulsed dye laser (PDL) (577, 585, and 595 nm) remains the gold standard for treating PWSs. However, it is estimated that less than 10% of patients will achieve complete clearance by using PDL [72]. According to previous studies, approximately 16% to 50% of patients’ lesions re-darken after treatment with PDL [73,74]. The recurrence of PWSs has been hypothesized to be associated with the induction of angiogenesis and the revascularization of lesions following the laser-induced destruction of dermal vessels.

PDL was found to be effective for vessels less than 400 μm in the dermis with a mean vessel diameter of 38 μm and were poorly effective for vessels with a mean vessel diameter of 19 ± 6.5 μm [75]. Similarly, another study revealed that optimal wavelengths for light-pigmented skin with small and shallow vessels were 580–610 nm in the visible band, which coincided with the wavelengths of PDL [76]. To analyze the vessel depth and diameter in different types of PWSs, our study included 35 patients with PWSs, of four distinct types, namely red PWSs, purple PWSs, hypertrophic PWSs, and nodular PWSs. The mean vessel diameters of the different types were 38.7 ± 5.9 μm, 93.5 ± 9.7 μm, 155.6 ± 21.8 μm, and 155.6 ± 29.54 μm, respectively. The mean vessel depths were 396.4 ± 31 μm, 944.2 ± 105.4 μm, 2971 ± 161.3 μm, and 3594 ± 364.6 μm, respectively [25]. PWSs were 38.7 ± 5.9 μm and 396.4 ± 31 μm, respectively, consistent with the best treatment conditions for PDL. Therefore, the potential curative effect of red PWSs was better than that of other types. Compared with the PDL, a 755 nm Alexandrite laser penetrates 50–75% deeper into the skin with less absorption of epidermal melanin [77]. Additionally, the penetration depth of a 1064 nm Nd: YAG laser is 4–7 mm into the dermis [78]. Although the treatments with 755 nm Alexandrite and 1064 nm Nd: YAG lasers for purple and hypertrophic PWSs showed high effectiveness, mild side effects were observed in half of the patients [18]. Moreover, CO_2_ lasers were found to be very useful for the treatment of the vascular nodularity of PWSs, and the large-spot 532 nm KTP laser with cryogenic spray also demonstrated promising results in treating PDL-resistant PWSs [79,80,81]. Although intense pulsed light (IPL) was effective in treating PDL-resistant PWSs, subsequent studies revealed that either IPL or IPL combined with PDL was not more effective than PDL alone in treating PWSs [82,83,84]. In recent years, photodynamic therapy (PDT) has been used as a treatment for PDL-resistant PWSs and large area PWSs. Almost half of the patients showed excellent or good levels of improvement (more than 50% color blanching) with few adverse events [11]. Although PDT is a promising treatment, half of the patients still do not achieve complete clearance.

### 3.2. Antiangiogenic Agents Combined with PDL

The significant recurrence rate after PDL treatment remains an important issue. Neoangiogenesis causes the recurrence of PWSs after laser treatment, which is mediated by the vascular endothelial growth factor (VEGF) pathway [85,86]. Gao, L. et al. also found that chemokine ligand 2 (Ccl2), matrix metalloproteinase 19 (MMP19), and plasminogen activator urokinase (Plau) were increased after PDL treatment [86]. Additionally, the overexpression of VEGF and its receptor (VEGF-R2) was found in PWSs, compared with healthy skin [87]. Axitinib, an oral antiangiogenesis agent, is known as the inhibitor of multi-receptor tyrosine kinases, including VEGFRs and PDGFR-beta [88]. Axitinib ointment effectively inhibited the PDL-induced increase in the mRNA levels of hypoxia-inducible factor (HIF)-1α, VEGF, Ccl2, MMP19, and Plau (Figure 2) [86]. Moreover, it also blocked PDL-induced pERK, pAKT, and pP70S6K. Meanwhile, previous studies revealed that ERK, AKT, and P70S6K were activated in PWSs. Thus, apart from treating PDL-induced angiogenesis, axitinib may also inhibit the progression of PWSs.

Beta-blockers (propranolol and timolol) are mainly utilized to treat infantile hemangiomas, which can prevent neovascularization by inhibiting VEGF, basic fibroblast growth factor (bFGF), MMP2, and MMP9 [89]. Hence, some scholars have tried to treat PWSs with PDL combined with topical beta-blockers to control the recurrence of PWSs. However, a multicenter randomized controlled trial revealed that the application of topical timolol gel for preventing neoangiogenesis did not significantly improve the recurrence of PWSs when treated with PDL [90]. *GNAQ* mutations are known to mildly activate the MAPK pathway, and they may stimulate the expression of endothelin, which could prevent neoangiogenesis. Thus, Taquin et al. used oral bosentan (2 mg/kg twice daily), an inhibitor of endothelin receptors, combined with PDL, to treat PDL-resistant PWSs in four patients [91]. Only one patient was markedly improved.

Imiquimod is a topical immune response modulator that is used to treat superficial basal cell carcinoma, actinic keratosis, and vascular tumors including infantile hemangiomas and Kaposi sarcoma.

The antiangiogenesis of imiquimod is for (1) the downregulation of proangiogenic factors including MMP-9 and bFGF and (2) the upregulation of antiangiogenic cytokines, including interleukin (IL)-10, IL-12, IL-18, interferon-alpha, and the tissue inhibitors of metalloproteinases [92]. According to a retrospective study of 20 patients, PDL combined with 5% imiquimod cream resulted in superior blanching responses, compared with PDL alone, in treating PWS lesions [93]. Similarly, another study also revealed that PDL combined with imiquimod 5% cream achieved greater color improvement than PDL and a placebo [94]. Moreover, the expression of VEGF and ANGPT2 decreased in the PDL + imiquimod sample compared with PDL alone, whereas the levels of MMP-9 and bFGF increased (Figure 2) [94]. To date, only two studies with small samples focused on treating PWSs by using PDL combined with imiquimod.

Rapamycin (RPM), a specific inhibitor of the mammalian target of rapamycin (mTOR), is an immunosuppressive and antiproliferative agent. RPM is already used against lymphangioma and lymphangiomatosis as insurance medical treatment in Japan. RPM has been reported to inhibit tumor growth by disrupting antiangiogenesis. The antiangiogenic properties of RPM might be achieved by reducing HIF-1α synthesis, which can decrease the expression of VEGF. Hence, RPM is used to treat vascular tumors and malformations, including angiomyolipoma, Kaposi sarcoma, and Kaposi hemangioma. Several studies evaluated the effect of RPM on neoangiogenesis after light-induced photothermolysis. These studies showed that RPM after PDL inhibited vessel reformation and perfusion, and RPM inhibited the PDL-induced expression of Ki-67, nestin, HIF-1α, VEGF, and P70S6K (Figure 2) [95]. Another study also demonstrated that RPM inhibited the increase in angiogenic factors, induced by PDL, through the AKT/mTOR/P70S6K pathway [96]. Meanwhile, previous studies revealed that AKT, ERK, and P70S6K were activated in PWS vessels [60].

A 37-year-old man with PWSs involving the left anterior chest was given 2 mg RPM orally daily for 7 days [97]. After PDL treatment, the patient was prescribed oral RPM for 4 weeks. Improved blanching was maintained for 13 months in PWSs treated with PDL combined with oral RPM compared with those treated with PDL alone. Similarly, in a 56-year-old male patient, the color and texture of PWSs treated with a combination of PDL and topical 0.5% RPM ointment significantly improved, compared with those treated with PDL alone [98]. In phase II, a randomized, double-blind clinical trial including 23 patients with SWS and facial PWSs revealed that the area treated with PDL + RPM (1% cream) showed a statistically significant improvement, compared with other interventions [99]. Case reports including five patients with PWSs also showed that the lesions treated with PDL and 0.5–1% topical RPM significantly improved over a short duration [100]. However, some studies found that the combination of PDL and RPM did not improve PWSs significantly, compared with PDL. A prospective, intrapatient, randomized controlled trial including 14 patients with PWSs mainly involving the trunk and extremities showed that the application of 0.1% RPM solution after PDL treatment did not improve PWS blanching [101]. Another study also demonstrated that topical 0.2% RPM as an adjuvant to PDL treatment did not improve PWS erythema [102]. Some scholars speculated that the cause of treatment failure was due to the low concentration of RPM, which was lower than the concentration in other studies (0.5% to 1%). However, another study evaluated the effect of the combination of 1% topical RPM with PDL, and only one in six patients showed improvement [103]. Overall, no statistically significant difference between the combination of 1% topical RPM with PDL and PDL alone was observed. Tixel drug delivery system (DDS) is a non-laser thermomechanical system that involves dehydrating the stratum corneum, which improves drug delivery by increasing skin permeability. To improve the treatment outcome, the DDS combined with 0.2% RPM cream and PDL were used to treat PWSs. The DDS combined with RPM improved the results of PDL treatment for PWSs and reduced the number of PDL sessions [104].

Although RPM can inhibit angiogenesis by reducing the expression of angiogenic factors after PDL treatment, the current clinical data remain ambiguous. The cause of the instability may be as follows: (1) fluctuations of the drug concentration, (2) the timing of pharmacological treatment, and (3) different types and locations of skin lesions. Thus, more prospective and large-scale investigations are needed to confirm these findings.

### 3.3. Future Perspectives in the Treatment of PWSs

Currently, PDL is the gold standard treatment of PWSs, but the recurrence rate is high. Although imiquimod and RPM combined with PDL reduced the recurrence rate in some studies, its clinical evidence is insufficient. Preliminary evidence shows that axitinib reduces the recurrence of PWSs in principle, but further clinical trials are needed to identify its efficacy and safety. Though adjuvant drugs can improve the recurrence rate after PDL treatment, they still cannot fundamentally resolve the problem of treating deep dermal vessels.

As PDT has developed rapidly in recent years, the effectiveness of PWS treatment has greatly increased. Copper vapor lasers and 532 nm LED green light are used to activate the photosensitizers, but the penetration depth of the laser and LED is limited [105,106]. A study revealed that PDT destroyed the vessels within 800 μm of the dermal–epidermal junction [107]. Moreover, the mean vessel depths of red, purple, and hypertrophic PWSs were 396.4 ± 31 μm, 944.2 ± 105.4 μm, and 2971 ± 161.3 μm, respectively. Hence, the treatment of hypertrophic PWSs remains difficult, and new PDT with next-generation photosensitizers is needed [105]. PDT requires three elements: light, photosensitizers, and molecular oxygen. Near-infrared fluorescence dyes, including indocyanine green (ICG), IR-780, IR-783, and IR-808, can generate strong fluorescence emissions in the range of 700 to 1000 nm [108]. These photosensitizers are effective in treating PWSs due to their depth of penetration. However, they have disadvantages such as poor stability and rapid decomposition in polar solutions, as well as low quantum yields. Nanocarriers, a new drug delivery system, are able to increase the bioavailability of photosensitizers [109]. Until now, PDT for treating PWSs has not been able to specifically target blood vessels. Previous studies have already shown that the ECs in PWS vessels have a specific phenotype of CD133^+^/CD166^+^/EphB1^+^/EfnB2^+^ [27]. Moreover, VEGFR-2 is overexpressed in PWS vessels [87]. Guo, X. et al. found that the nanoparticles with CD133 aptamers and propranolol significantly inhibited infantile hemangioma [110]. Hence, using photosensitizers that are conjugated nanoparticles and surface markers (monoclonal antibodies and peptides) possesses good therapeutic potential in treating PWSs (Figure 2).

In recent years, photothermal therapy (PTT), a novel non-invasive therapeutic technology, has become attractive for cancer treatment. The photothermal transduction agents (PTAs) harvest the energy from the light of a specific wavelength and convert it into heat, thereby increasing the temperature of the surrounding microenvironment [111]. Cells would die rapidly at 46–52 °C due to microvascular thrombosis and ischemia [112]. PTAs can be divided into inorganic materials (graphene, carbon nanotubes, boron nitride) and organic materials (NIR-responsive small molecules) [111]. Typically, PTA absorptions are adjusted between 750 and 1350 nm to match the tissue-transparent window. Therefore, some scholars try to use PTT to treat PWSs due to the possibility that PPT has therapeutic effects on deep vessels. A study revealed that blood absorbance increased 3.9 times at 1064 nm after the injection of gold PEG-modified gold nanorods into the blood, and the treatment threshold energy density for PWSs decreased by 33% [113]. In another study, compared with the vessels without NIR erythrocyte-derived transducers irradiated at 585 nm, the vessels containing NIR erythrocyte-derived transducers showed higher levels of photothermal damage [114]. The above studies show that PPT may have the potential to treat PWSs. For improved vascular targeting of PTAs, PTAs should be combined with surface markers to treat PWSs.

## 4. Summary and Prospects

PWSs are congenital progressive vascular malformations involving the skin and mucosa. To date, the factors involved in disease progression and pathogenesis in PWSs are yet unclear. The cause of dilated blood vessels is suggested to be related to the following factors: (1) somatic *GNAQ* (R183Q) mutations, with a mutation frequency of 55.6% to 100%, enlarged blood vessels that resemble capillary malformations by ANGPT2, (2) the decrease in perivascular nerve elements, altering the neural modulation of vascular tone, thus affecting progressive vascular ectasia, (3) the coexistence of Eph receptor B1 and ephrin B2, which disrupts normal EC–EC interactions, resulting in progressive vessel dilation, and (4) the abnormal αSMA of pericytes contributes to the progressive dilatation of PWS blood vessels. In addition, the development of PWSs is related to the aberrant activation of ERK, c-JNK, P70S6K AKT, PI3K, and PKCα. Though some of these kinases are activated by the somatic *GNAQ* (R183Q) mutations, the reason for activating the remaining kinases, including P70S6K, PLC-γ, and PKCα, remains unclear. Moreover, the relationship between *GNAQ* (R183Q) mutations and nerve defects is uncertain. Although PDL remains the gold standard of PWS treatment, the recurrence rate is high. Interestingly, topical drugs such as imiquimod, axitinib, and RPM, combined with PDL treatments, may alter recurrence rates and reduce PDL sessions for PWSs. For the deep vascular plexus, the utilization of photosensitizers or PTAs encapsulated by nanocarriers conjugated to surface markers (monoclonal antibodies and peptides) possesses good therapeutic potential in PDT or PTT for PWSs. In conclusion, the pathogenesis, progression, and treatment of PWSs still require extensive investigation.

## Figures and Tables

**Figure 1 ijms-23-12139-f001:**
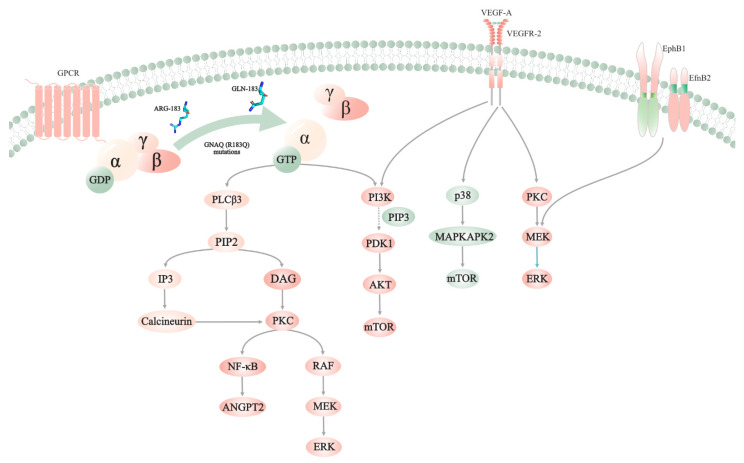
The potential pathogenesis of PWSs. *GNAQ* (R183Q) mutations activate classic MAPK, PKC/NF-κB/ANGPT2, and PI3K/AKT/mTOR pathways. Meanwhile, overexpression of VEGF-A and VEGFR2 activates PI3K/AKT/mTOR, p38-MAPK/mTOR, and classic MAPK signaling pathways. Furthermore, the coexpression of EphB1/EfnB2 activates the classic MAPK pathway. Activating these signaling pathways could influence the proliferation, migration, permeability, and survival of cells, thereby promoting angiogenesis and the dilation of vessels.

**Figure 2 ijms-23-12139-f002:**
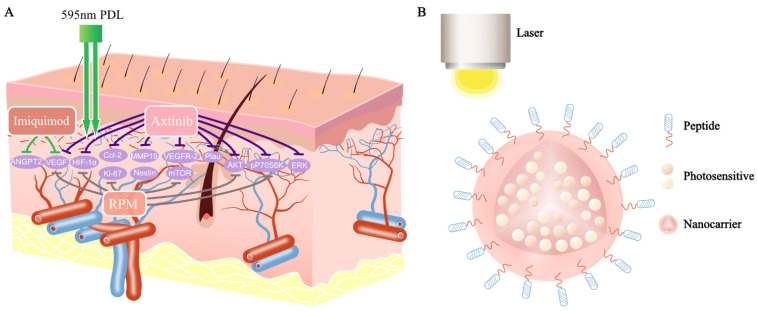
Antiangiogenic agents combined with PDL and a novel PDT for treatment of PWSs: (**A**) axitinib inhibits the mRNA levels of HIF-1α, VEGF, Ccl2, Plau, and MMP19 increased by PDL. In addition to this, it also blocks PDL-induced pERK, pAKT, and pP70S6K. The expression of VEGF and ANGPT2 decreased in the PDL combined with the imiquimod sample, compared with PDL alone. RPM inhibits the PDL-induced expression of Ki-67, nestin, HIF-1α, VEGF, P70S6K, etc.; (**B**) a novel PDT for treatment of PWSs: photosensitizers (ICG, IR-780, IR-783) are encapsulated inside nanocarriers, which are conjugated with special markers (CD133/CD166/EphB1/EfnB2/VEGFR-2).

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
