# Peer review of "Pathogenesis of Port-Wine Stains: Directions for Future Therapies"

_ijms, 2022, doi:10.3390/ijms232012139_

Round 1

Reviewer 1 Report

Congratulations for your review.

comment: when you say “PWS could develop into VM,ACM,PG and cavernous vascular extasía” ishoul ve change to “A hypothesis could be that PWS…”

Regarding to laser treatment, it should be added that co12 laser is very usefull for the treatment of vascular nodularity of pws. Also new KTP 532 10 mm spot with cryogenic spray could bring promisión results in PDL RESISTANT PWS

Author Response

(1) Comment: when you say “PWS could develop into VM,ACM,PG and cavernous vascular extasía” ishoul ve change to “A hypothesis could be that PWS…”

Thank you for your useful suggestions on our manuscript very much. Based on your suggestion and discussion from our team, we believe this statement as following is more rigorous.

Line 157 - 159

To summarize, PWS are initially characterized as capillaries and/or postcapillary venous malformations, and nodules in the PWS can be VM, AVM, PG, or cavernous vascular ectasia.

(2) Regarding to laser treatment, it should be added that co12 laser is very usefull for the treatment of vascular nodularity of pws. Also new KTP 532 10 mm spot with cryogenic spray could bring promisión results in PDL RESISTANT PWS

In accordance with your useful suggestions, we have added this section

Line 366-369

Moreover, the CO2 laser was very useful for the treatment of vascular nodularity of PWS, and the large spot 532 nm KTP laser with cryogenic spray also demonstrated promising results in treating PDL-resistant PWS[79-81].

Reviewer 2 Report

Comprehensive review on the pathogenesis and therapeutic approaches of port wine stains. The manuscript is well organized and clear. Figure 3 should be improved. Word size (drugs and kinases) is small and panel B of little impact. The main concepts are reported and briefly presented. The paragraph on Gprotein can be shortened, however the part relating to future approaches and treatments under study should be updated. The fluency of the reading of the text is to be improved, in general the manuscript requires a moderate revision of the English.

Author Response

  • Figure 3 should be improved. Word size (drugs and kinases) is small and panel B of little impact.

Thank you for your useful suggestions on our manuscript very much. We modified the Figure 3

  • The paragraph on G protein can be shortened, however the part relating to future approaches and treatments under study should be updated.

Thank you for your useful suggestions very much. We made the following changes:

  1. We shortened the paragraph on G protein(Guanine nucleotide-binding proteins (G proteins) are heterotrimers composed of 3 subunits: α, β, and γ[47]. Gα subunits are differentiated into four families, Gs, Gi, Gq, and G12[48]. Moreover, the Gq family consists of Gαq, Gα11, Gα14, Gα15, and Gα16.)
  2. We updated future approaches and treatments under study about PWS.

Line 366 – 371

Moreover, the CO2 laser was very useful for the treatment of vascular nodularity of PWS, and the large spot 532 nm KTP laser with cryogenic spray also demonstrated promising results in treating PDL-resistant PWS[79-81]. Although intense pulsed light (IPL) was effective in treating PDL-resistant PWS, subsequent studies revealed that either IPL or IPL combined with PDL was not more effective than PDL alone in treating PWS[82-84].

 Line 490 – 560

In recent years, photothermal therapy (PTT), a novel noninvasive therapeutic technology, has become attractive for cancer treatment. The photothermal transduction agents (PTAs) harvest the energy from the light of a specific wavelength and convert it into heat, thereby increasing the temperature of the surrounding microenvironment[111]. Cells would die rapidly at 46–52 °C due to microvascular thrombosis and ischemia[112]. PTAs can be divided into inorganic materials (graphene, carbon nanotubes, boron nitride) and organic materials (NIR-responsive small molecules)[111]. Typically, PTAs absorp-tions are adjusted between 750 and 1350 nm to match the tissue-transparent window. Therefore, some scholars try to use PTT to treat PWS for the possibility that PPT has therapeutic effects on the deep vessels. A study revealed that blood absorbance increased 3.9 times at 1064 nm after injecting of gold PEG-modified gold nanorods into the blood, and the treatment threshold energy density for PWS decreased by 33%[113]. In another study, as compared with vessels without NIR erythrocyte-derived transducers irradiated at 585 nm, vessels containing NIR erythrocyte-derived transducers showed higher levels of photothermal damage[114]. The above studies show that PPT may have the potential to treat PWS. For improved vascular targeting of PTAs, PTAs should be combined with surface markers to treat PWS.

  • The fluency of the reading of the text is to be improved, in general the manuscript requires a moderate revision of the English.

Thank you for your useful suggestions. The English language has been polished by a professional editor from MJEditor (www.mjeditor.com).

Reviewer 3 Report

I think this report is an excellent review of a wide range of literature, including valuable reports on PWS. However, in order to publish this report, some corrections are required.

Comment 1

Line 11 - 12As the disease progresses, capillary malformations (CM) develop into venous and arteriovenous malformations”

Line79 – 86, line 484 - 485

I feel uncomfortable with the above statement that capillary malformations could develop into venous malformations and arteriovenous malformations.

However, I agree with the following two points.

Over 50 years, the tissue within the nodule has no pathologic evidence of a simple capillary malformation.

Because GNAQ mutations influence PI3K/AKT/mTOR pathway and p38-MAPK/mTOR pathway, venous malformation-like findings within the nodules of PWS may be observed microscopically over time.

On the other hand, physicians must diagnose patients with capillary vascular malformations, lymphatic malformations, venous malformations, and arteriovenous malformations by comprehensively reviewing clinical, imaging, and pathological findings. Diagnosis based solely on pathological findings has a high risk of being misdiagnosed.

If you quote in this report their conclusion that nodular capillary malformations transform into venous or arteriovenous malformations in the cited literature, it is to be expected that readers will confuse tissue changes less than 2 cm in diameter with the change of patients` original disease. Since this report is a review, it is necessary to emphasize that the affected vessel changes are at the microscopic level. It may be necessary to show that the absence of RASA1 gene mutations in childhood PWS skin lesions and the presence of frequent RSASA1 gene mutations in PWS nodular lesions in order to conclude that nodular capillary malformations can transform into arteriovenous malformations at this time in 2022.

Comment 2

Line 58 - 63

In some classifications of PWS, if there is a representative classification that is directly linked to treatment selection against PWS, please describe it in in your report.

Depth and mean diameter of the affected vessels are described in the treatment chapter of PWS (line 353 -355, line 458- 459). If there is a relationship between the classification of PWS and these pathological or anatomical analyses, please describe them.

Comment 3

Line 158 -163

It is difficult to understand the meanings and relationships of 2, 4, and 1. A diagram showing the family relationships in the cited report would be helpful to understand the family case. I suggest to modify the diagram slightly and include it in this report.

Comment 4

Line 180

Table 1 is missing. Is it because of an editor's mistake? Please confirm table 1.

Comment 5

Line186

Is it need to be abbreviated as NBS? After this, NBS appears only once in the report.

Comment 6

Line 249 - 250

I wonder that SMARCA4 is SMARCA4? or BRG1 is BRG1?

Comment 7

Line282

What does WES stand for?

Comment 7

Line 371 - 372

Are the control specimens non-irradiated skin of PWS? Is it healthy skin?

Comment 8

Line 408 - 409

In Japan, RPM is already used against lymphangioma and lymphangiomatosis as insurance medical treatment. Please add this.

Comment 9

Line437

Tixel doesn't seem to be popular yet. Please provide additional explanation about Tixel.

Does Tixel lead subcutaneous bleeding by damaging dilated blood vessels in the affected skin? 

Author Response

Thank you for your useful suggestions.

Round 2

Reviewer 3 Report

Thank you for correcting the manuscript.

I accept that this review will be published.